# Transmission of Methicillin-Resistant *Staphylococcus* spp. from Infected Dogs to the Home Environment and Owners

**DOI:** 10.3390/antibiotics11050637

**Published:** 2022-05-10

**Authors:** Mari Røken, Stanislav Iakhno, Anita Haug Haaland, Yngvild Wasteson, Ane Mohn Bjelland

**Affiliations:** 1Department of Paraclinical Sciences, Faculty of Veterinary Medicine, Norwegian University of Life Sciences, 1433 Ås, Norway; yngvild.wasteson@nmbu.no (Y.W.); ane.mohn.bjelland@nmbu.no (A.M.B.); 2Research and Development, Previwo AS, 0454 Oslo, Norway; stanislav.iakhno@previwo.no; 3Department of Companion Animal Clinical Sciences, Faculty of Veterinary Medicine, Norwegian University of Life Sciences, 1433 Ås, Norway; anita.haug.haaland@nmbu.no

**Keywords:** antimicrobial resistance, methicillin-resistance, one health, *Staphylococcus pseudintermedius*, *Staphylococcus epidermidis*

## Abstract

Dogs with methicillin-resistant *Staphylococcus* spp. (MRS) infections often undergo treatment in their homes, interacting with their owners and surroundings. This close contact between dogs and owners may facilitate the interspecies transmission of MRS. Therefore, this study aimed to investigate the transmission of MRS from infected dogs to their owners and home environments. Seven households with dogs that had been diagnosed with methicillin-resistant *S. pseudintermedius* (MRSP) and one household with a dog with methicillin-resistant *S. epidermidis* (MRSE) participated in the study. Dogs, owners, and the home environments were screened for the presence of clinical MRS. A selection of 36 staphylococcal isolates were whole-genome sequenced and screened for resistance genes and virulence genes. Clinical MRS were primarily identified from the dogs and their immediate surroundings, but these were also detected in locations that were out of reach for the dogs, indicating indirect transmission. Two of eight owners carried clinical MRS in their nostrils, while one owner carried methicillin-susceptible *S. pseudintermedius* (MSSP). All clinical MRS were multi-resistant, and several possessed resistance genes that were not expressed phenotypically. Clinical MRSP persisted in the home environment for a prolonged period, despite infection recovery and one dog being euthanized. Regardless of the stable presence of MRSP in the surroundings, the owners in these homes remained negative, but tested positive for MSSP on three occasions.

## 1. Introduction

Methicillin-resistant *Staphylococcus* spp. (MRS) cause a substantial number of infections in humans worldwide. Methicillin-resistant *Staphylococcus aureus* (MRSA) has been estimated to cause almost 150,000 infections annually, and over 7000 attributable deaths in the European Union and the European Economic area [1]. Methicillin-resistant *Staphylococcus pseudintermedius* (MRSP) is among the most common MRS carried by and causing infections in dogs [2], and it is the canine equivalent to *S. aureus*. Despite initially being described as an animal pathogen, an increasing number of studies now recognize *S. pseudintermedius* as an opportunistic human pathogen [3,4,5,6]. In human medicine, methicillin-resistant coagulase negative *Staphylococcus* spp. (MRCoNS), and methicillin-resistant *S. epidermidis* (MRSE) are major contributors to nosocomial infections [7,8,9]. Similarly, in veterinary medicine, MRCoNS are recognized to colonize and cause infections in dogs, with MRSE being one of the most commonly occurring MRCoNS species [10,11,12]. In addition to *mecA*-encoded resistance to all beta-lactam antibiotics, clinical strains of MRS often have multidrug-resistant properties that complicate the treatment of these infections [13,14].

The close relationship between dogs and humans may facilitate the bidirectional transmission of bacteria. Transmission may occur through direct contact and/or indirectly through contact with bacteria in the surrounding home environment. The staphylococci’s ability to survive without a host from weeks to months in dry environments increases the probability of MRS exposure and allows for the recolonization of hosts after successful antimicrobial treatment of the primary infection [15,16].

As clinical microbiologists and veterinarians, we are often contacted by dog owners who worry about the risk of becoming infected by their dogs. Therefore, this study aimed to investigate the transmission of clinical MRS from dogs to their immediate surroundings. By screening the dogs, their owners, and home environments for clinical MRS, we assessed the transmission potential of MRSP and MRSE. Furthermore, we aimed to describe the MRS’ resistome and virulence genes to evaluate the severity of zoonotic transmission.

## 2. Results

### 2.1. Identification of MRS Isolates

An extended summary of all MRSP, MSSP, and MRSE isolates included in the study is presented in Appendix A. A total of 103 isolates were included, 62 from Sampling 1, and 41 from the follow-up samplings (Sampling 2 and 3). Table 1, Table 2, Table 3 and Table 4 present summaries of the data from Appendix A.

### 2.2. Location of Clinical Methicillin-Resistant Staphylococci

Overall, the results from Sampling 1 showed that clinical MRS were frequently present on the dogs’ carrier sites (perineum/mouth) and in the home environment (Table 1). Two of eight owners carried the same MRS that their dogs were infected with, one case of MRSP (Household A), and one case of MRSE (Household H). In both cases, the isolates were recovered from the owners’ nostrils. In addition, the owner of household G carried MSSP in the nose. Except for Dog B, all of the dogs tested positive for clinical MRS at either one or two carrier sites. We detected clinical MRS in all home environments, but in varying locations and frequencies. In all households, the MRS were identified at a minimum of one of the dog-associated locations; the food bowl, the sleeping place, or the floor, while we could identify the MRS in four of the kitchens and two of the bathrooms.

### 2.3. Contact Dogs

Contact dogs were present in three households. Despite having tested positive for MRSP in a screening a month before, the contact dog in household C tested negative for MRSP at the sampling for this study. The contact dog in household F tested positive for MRSP from the perineum and from pyotraumatic dermatitis on the cheek. Dog E had 10 four-week-old puppies that all tested positive for MRSP.

### 2.4. Phenotypic Resistance

All MRS isolates were multidrug-resistant by the definition proposed by Magiorakos et al. [17]. The number of resistance classes ranged from three to seven, with the MRSP isolates in households B and C expressing phenotypic resistance to most classes of antibiotics (Table 2). Resistance to erythromycin, trimethoprim/sulfamethoxazole, clindamycin, and tetracycline were the most frequent. None of the MRS were resistant to chloramphenicol, while the MRSE isolate was the only isolate expressing resistance to fusidic acid. Of the eight MRSP isolates in household A, three were susceptible, while the remaining were resistant to clindamycin. The MSSP isolated from the owner in household G was susceptible to oxacillin, but had an otherwise identical resistance pattern to the MRSP isolated from the dog and the home environment.

### 2.5. Genomic Data Analysis

Table 4 presents the staphylococcal cassette chromosome *mec* (SCC*mec*) elements, STs, and antimicrobial resistance genes of clinical MRS from Sampling 1. SCC*mec* IVg (2B) was the most frequent SCC*mec* element in the MRSP, being detected in four of seven isolates. Three of seven MRSP were typed to ST258. This ST was shared by the MRSP and MSSP isolates from household G (Appendix A). Furthermore, the MRSP and MSSP isolates carried the same resistance genes, except for the *mecA* gene.

Overall, the genotypic resistance corresponded well with the phenotypic resistance, with some exceptions: A broad spectrum of aminoglycoside resistance genes were present in all MRSP isolates, except for Household A. In Household A, none of the MRSP isolates expressed phenotypic resistance to trimethoprim/sulfamethoxazole, while the resistome analysis uncovered the *dfrG* gene in all of the sequenced isolates. Furthermore, despite their phenotypic heterogenic resistance to clindamycin, all of the MRSP isolates from Household A possessed the *ermB* gene. A Blast analysis revealed a C251T mutation (Ser84Leu) in the *gyrA* genes of the fluoroquinolone-resistant MRSP isolates in Households B–D. The MRSE isolates were susceptible to enrofloxacin despite possessing *norA*, a gene encoding a multidrug efflux pump conferring resistance to fluoroquinolones. As with *norA*, the trimethoprim resistance gene *dfrC* was present in the MRSE genomes but this was not expressed phenotypically.

### 2.6. Persistence over Time

Households A and B were sampled for two periods of five days. During both sampling periods, Dog A displayed infection symptoms and tested positive for MRSP until the sampling was terminated, while the home environment was intermittently positive (Table 4). The owner tested negative for MRSP during both sampling periods, but tested positive for MSSP on one occasion. The MSSP isolates were phenotypically susceptible to all antibiotics included in the panel, and the resistome analysis confirmed the absence of resistance genes. The MSSP sequence type could not be established using multilocus sequence typing (MLST).

The situation in Household B differed from Household A. The dog displayed no symptoms of infection on the first day of the follow-up sampling. It tested positive for MRSP from the primary infection site on the first day, but remained negative on the following test days. With one exception, in Sampling period 3, the perineal and mouth samples were negative for MRSP. Dog B’s samples were dominated by an MSSP strain that we also isolated from the owner on two occasions, one in Sampling Period 2 and one in Sampling Period 3. On both occasions, the owner tested negative for MSSP the following day. MLST could not establish the MSSP sequence types. The isolates expressed no phenotypic resistance to the antibiotics in the test panel, and we detected no resistance genes in the search against the CARD database. Despite Dog B’s recovery from the infection and the negative carrier status, the home environment remained positive for MRSP throughout the testing period.

Household C was sampled 5 and 10 weeks after the dog had been euthanized. The home environment tested positive for MRSP, with two different phenotypic resistance patterns in the first follow-up sampling. One isolate expressed the same phenotypic susceptibility profile as the MRSP recovered from the dog six weeks earlier (T/S, Tet, Enr, Gen, Cli, Oxa, and Ery). In contrast, the other isolate was susceptible to trimethoprim/sulfamethoxazole, gentamicin, and erythromycin. The CARD analysis showed that the less resistant isolate lacked *ant(6′)*, *aph(3′)*, *dfrG*, *ermB*, and *sat4*. Like the isolate recovered from the dog, we could not determine the STs by MLST on either of the two environmental MRSP isolates. The SCC*mec* elements were identical (V) to the MRSP isolated from the dog from Sampling 1. In addition, the virulome analysis revealed that both isolates had identical virulence genes (Appendix A). No MRSP was detected in the second follow-up sampling 10 weeks after Dog C was euthanized. The owner tested negative for MRSP on both follow-up occasions.

### 2.7. Virulence

All MRSP/ MSSP isolates possessed genes that are involved in adhesion and biofilm production, *ebpS* and *icaA-D*, and the toxin-encoding genes *hlB*, *lukF-I*, *lukS-I*, *se-int*, *siet,* and *speta* (Appendix A). Except for MRSP from Households B and D, all isolates had the adhesin gene *spsD*. Instead, MRSP from Households B and D possessed another adhesion gene, *spsL*. The MSSP isolates from Households A and B possessed the bacteriocin-encoding gene *bacSp222* and the enterotoxin-encoding gene *sec3*, which are unique to these strains. In addition to the MRSP isolate from Household F, these were the only isolates in possession of the exfoliative toxin-encoding gene *expB* and the surface protein-encoding gene *spsI*. Compared to the MRSP isolate from Household G, the MSSP isolate from the same household lacked the *mecA* gene and the surface protein-encoding genes *spsG* and *spsM*.

Similar to the MRSP/MSSP isolates, the MRSE isolates from Household H possessed a rich variety of virulence genes, including genes encoding adhesins; *aae*, *atlE*, *bhp*, *ebpS*, *fbe*, *gehC*, *gehD*, and *sdrF-H.* Genes involved in the regulation of biofilm production, *htrA*, *sepA*, and *sspA*, were present, but the biofilm-producing genes *icaA-D* were not identified.

## 3. Discussion

An increasing number of reports state that *S. pseudintermedius* is an opportunistic human pathogen, while *S. epidermidis* can cause infections in several species [18,19,20,21,22,23]. Considering the close relationship between dogs and owners, we aimed to investigate the transmission of MRS from clinical cases in dogs. By analyzing their locations, the antimicrobial resistance properties, and the virulence genes of MRS, we assessed the transmission of MRS to the surroundings, and the severity of potential zoonotic transmission to owners.

The results indicate that clinical MRS are primarily located on the dogs and in their immediate surroundings. Household F was the exception, but a plausible explanation could be that Dog F’s movement was confined to an enclosure in the living room. Unlike the other participating dogs, Dog F’s infection site was covered by bandages, thus limiting bacterial shedding. Clinical MRS were present in locations that were out of reach for the dogs in half of the households, indicating an indirect transmission route, either by dust particles or mechanical vectors such as cleaning cloths or hands [24,25]. We detected one case of MRSP and one case of MRSE among the owners, both isolated from the owners’ nostrils. In the case of MRSP, we can be reasonably certain that the MRSP had been transmitted from the dog to the owner, as it is primarily a canine-associated bacteria. In the case of MRSE, however, the transmission route is less clear. *S. epidermidis* has a broad spectrum of mammalian hosts, including dogs and humans [26]. ST640 has previously been reported in humans and dairy cows, but not in dogs [27,28]. MSSP was likely the primary cause of Dog H’s infection, as it is a common bacteria that is isolated from canine pyotraumatic dermatitis [29]. However, we cannot exclude the possibility of the opposite, as no results from previous bacterial culturing were available. Regardless of which bacteria were the primary cause, MRSE was recovered from the perineum of the dog, indicating that the finding of MRSE from the infection site was not temporary contamination.

To better understand the dynamics over time, we continued with follow-up sampling for two periods in Households A and B. Dog B recovered from acute otitis externa at the beginning of Sampling 2. Except on one occasion, we did not detect MRSP from Dog B during Samplings 2 and 3. The absence of MRSP in Dog B could be due to the method’s detection limit. However, the isolation protocol included both an enrichment- and a selective culturing step, thus increasing the method’s sensitivity. Interestingly, Dog B’s samples were dominated by an MSSP strain that possessed the gene encoding the BacSp222 peptide. BacSp222 functions as a bacteriocin that kills Gram-positive bacteria, including related staphylococci [30]. Thus, it is tempting to hypothesize that the domination of MSSP prevented the colonization of MRSP. The dominating strain of MSSP was recovered from the owner’s throat/pharyngeal samples on two occasions, but it was not detected over the following days, indicating that the findings were temporary contamination.

As opposed to Dog B, Dog A presented with a more chronic clinical state, with both active and recovering dermal lesions throughout the sampling periods. Consequently, we recovered MRSP from the dog and the home environment throughout the sampling periods. Interestingly, an MSSP strain containing *bacSp222* was also detected twice in owner A. On both occasions, the MSSP could not be recovered the following day, thus supporting the theory that the MSSP was a temporary contaminant.

Despite the MRSP’s stable presence over time in Households A and B, the owners remained negative, indicating a species barrier. As the MRSP were selectively enriched, the bacterial load in the home environment could not be quantified. However, as Dog A continuously shed MRSP from the active lesions, we assume the quantity was higher than negligible. Consequently, the owner was continuously exposed to MRSP.

As exemplified in Households B and C, staphylococci can survive in nutrient-poor, dry conditions for weeks to months [31]. The home environment remained positive regardless of Dog B’s negative carrier state and the maintenance of regular house cleaning routines until sampling was terminated. It is likely that the MRSP detected in the home environment for the remaining sampling period originated from the initial infection. Dog C had been euthanized shortly after Sampling 1, and the contact dog was no longer present in the household. In the meantime, the owner had implemented several hygienic measures, but a thorough inspection revealed dog hair in various locations. Consequently, we recovered MRSP from the floor and the sofa five weeks after the dog was euthanized. In contrast, we could not detect MRSP from any environmental sample 10 weeks after euthanization, even though dog hairs still were present in the home environment. Hence, the MRSP had been eliminated or reduced to quantities that were below the detection limit sometime between 5 and 10 weeks after the dog was euthanized. Considering MRSP’s resilience and that it is easily transmitted between dogs [25], caution should be taken when introducing MRSP-naïve dogs to home environments that have been previously occupied by an MRSP-infected dog.

Though not a virulence factor, antimicrobial resistance genes offer a competitive advantage for bacteria when they are exposed to antibiotics. The phenotypic resistance analysis established that all the MRSP/MRSE isolates were multi-resistant by the definition proposed by Magiorakos et al. [17]. However, the genetic resistome analysis revealed some MRS carried resistance genes that were not apparent through phenotypic susceptibility testing. This was especially evident for the aminoglycoside resistance genes. The *ant(6′)-Ia*, *aph(3′)-IIIa*, *ant(4′)Ib*, *ant(9)-Ia*, and *sat4* genes encode proteins that are unaffected by gentamicin, the only aminoglycoside antibiotic included in the phenotypic panel. Furthermore, the resistome analysis revealed genes that were not expressed in vitro, including genes encoding trimethoprim- and clindamycin resistance, and the multidrug efflux pump NorA. These findings show that the antimicrobial resistance potential can be underestimated by relying on limited phenotypic susceptibility profiles alone.

Knowledge about *S. pseudintermedius* pathogenesis is still sparse [32]. Overall, we observed few differences in virulence genes among the MRSP and MSSP isolates. Virulence genes associated with adherence to host tissue such as *ebpS* and *lip*, the biofilm-associated genes *icaA-D*, and genes encoding the cytotoxins *lukF-I* and *lukS-I* were present in all of the sequenced isolates. In addition, most MRSP/MSSP isolates possessed *spsD*, the protein of which contains an A domain that is homologous to fibronectin-binding proteins and clumping factors, which are both important adhesins in *S. aureus* [33]. Furthermore, SpsD mediates the adherence to human fibronectin and is associated with the internalization of human osteoblasts in vitro [34]. In contrast to other more virulent staphylococci, such as *S. aureus, S. epidermidis* does not possess aggressive virulence properties [7]. As a well-adapted skin commensal, *S. epidermidis* has an arsenal of adhesins that enable it to maintain this lifestyle. The MRSE isolates in this study were no exception, which likely contributed to their ability to colonize both the owner and the dog in Household H.

## 4. Conclusions

This study has documented that the home environment is an important reservoir for clinical multidrug-resistant MRS that is shed by infected dogs. The locations in direct contact with the infected dogs were most frequently positive for clinical MRS. These locations stayed positive over an extended period, despite infection recovery, cleaning measures, and the absence of dogs. Hence, the human household members are exposed to clinical MRS directly through contact with the dogs, and indirectly through the home environment. The significance of this exposure is debatable. Undoubtedly, MRSP and MSSP can transmit from dogs to humans. However, the findings in this study and previous studies indicate that human carriership is rare and temporary [24,35]. MRSP and MSSP produce a broad range of virulence factors. Yet, many of the virulence factors have not been characterized. Given that a significant part of reported MRSP/MSSP infections in humans has been observed in patients with underlying diseases, host factors such as age and health state seem to be important [5,36]. In the MRSE-positive household, the transmission direction was not clear. Nonetheless, co-carriership in the dog and owner, and the vast presence of MRSE in the home environment indicate that MRSE transmit between dogs, humans, and the environment. Prophylactic measures to reduce the transmission risk of MRS could be considered for implementation in households in which immunocompromised individuals are exposed.

## 5. Materials and Methods

### 5.1. Participants

Dogs and their owners were recruited to the project from small animal clinics in the surrounding areas of Oslo. Dogs that had recently been diagnosed with an infection from which MRS could be cultured were included. All participants signed individual consent forms and answered a questionnaire regarding their dogs, professions, antimicrobial consumption, and travel habits. Eight households (A–H) participated in the study, of which seven were households with dogs with MRSP infections. The remaining dog had a co-infection with MRSE and MSSP. In addition to the infected dogs, one owner and eventual contact dog(s) from each household were included in the study. A summary of the participating dogs is presented in Table 5. An extended summary of the dogs and the participating households is presented in Appendix A.

### 5.2. Sampling

The samples were collected during the period from January 2020 to November 2021. Samples of the infection site, oral mucosa, and perineal samples were collected from the infected dogs using nylon flocked swabs (Eswab™ 480C, Copan group, Brescia, Italy). Samples were taken from the perineum and the oral mucosa from the contact dogs by a veterinarian. According to the veterinarian’s instructions, the owners collected swab samples from their nostrils and throat. The home environment was sampled using moist cloths (Sodibox^®^ Swab cloth, Nevez, France). Samples were collected from the pets’ food bowl and sleeping place, floor (living room and kitchen), bathroom (sink faucet and hand towel), and the kitchen (kitchen counter, dish towel, cloth, and sink faucet). The samples from the two latter locations were taken in areas that were out of reach from the pets. All households were sampled once.

Households A, B, and C participated in further sampling, as outlined in Figure 1. Households A and B were sampled over two periods for five subsequent days, with a four-week break in between. The first follow-up sampling was performed two weeks after Sampling 1. Both households were told to maintain their regular cleaning routines during this period. Only the floor, bathroom, and kitchen were included for the follow-up environmental samples.

The dog in Household C was euthanized approximately one week after Sampling 1. The owner and home environment were sampled 5 and 10 weeks after the dog was euthanized. Before the first follow-up sampling, the owner had cleaned and disinfected the floor with a disinfecting agent containing 58% ethanol and 0.1% alkyl dimethyl benzyl ammonium saccharinate. The carpets and curtains had been dry cleaned, and the dog bed had been removed. The environmental samples were then taken from the floor, sofa, bathroom, and kitchen.

### 5.3. Culturing and Species Identification

Swabs and cloths were analyzed individually. The swabs were vortexed for a minimum of 10 s before 10 µL was plated on 5% bovine blood agar and incubated overnight at 37 °C. Additionally, 100 µL of the liquid Amies were transferred to 7 mL of Mueller Hinton broth containing 6.5% NaCl. One hundred milliliters of MH broth was added to each cloth. All samples were incubated overnight at 35 °C before 20 µL of the MH broth was inoculated on Oxacillin Resistance Screening Agar Base (ORSAB, Oxoid, Basingstoke, Hampshire, UK) supplemented with 2 mg/L of oxacillin and incubated for 24 h at 35 °C. In cases of no growth after 24 h, the plates were re-incubated for 24 h before reading.

Presumptive staphylococcal colonies growing on ORSAB plates were subcultured on 5% bovine blood agar overnight and identified to the species level by using a combination of standard laboratory techniques such as colony morphology, tests for coagulase, catalase, ONPG, mannitol, and Matrix-assisted laser desorption/ionization time of flight (MALDI-TOF) (VITEK^®^ MS, bioMérieux, Craponne, France).

### 5.4. Susceptibility Testing

Verified staphylococcal isolates were susceptibility tested according to CLSI guidelines against 10 antibiotics, using the disk diffusion method (Rosco Diagnostica, Taastrup, Denmark). The panel consisted of: Trimethoprim + sulfa (1.25/23.75 µg), tetracycline (30 µg), fucidic acid (100 µg), enrofloxacin (5 µg), gentamicin (10 µg), clindamycin (2 µg), oxacillin (1 µg), cefoxitin (30 µg, MRSE only), chloramphenicol (30 µg), and erythromycin (15 µg). Phenotypically oxacillin/cefoxitin-resistant isolates were confirmed as being methicillin-resistant by *mecA* PCR [37]. Isolates were evaluated for multidrug resistance using the definition proposed by Magiorakos et al. and Sweeney et al. [17,38]

### 5.5. DNA Extraction and Whole-Genome Sequencing

We selected a subset of 36 MRS and MSSP from the different households for whole-genome sequencing (Appendix A). DNA extraction was performed using a modified version of the MasterPure™ Gram Positive DNA Purification Kit protocol (Appendix B) (Lucigen Corporation, Middleton, WI, USA). The DNA quality control and quantification were performed using a NanoDrop^®^ ND-1000 (ThermoScientific, Wilmington, CA, USA) and Qubit fluorometer with the dsDNA Broad Range Assay kit (Invitrogen, Eugene, OR, USA), respectively. The Norwegian Sequencing Centre (NSC) (Oslo, Norway) performed the library prep in two batches using the Swift Turbo 2S flex DNA library prep and Nextera DNA Flex prep protocols for Batches one and two, respectively. The change in the protocol was due to the Swift Turbo 2S flex prep having been phased out. The paired-end sequencing reads (300 bp) were obtained using the Illumina MiSeq platform v3 (NSC).

### 5.6. Bioinformatical Analysis

The raw sequencing reads were processed by adapter clipping and quality trimming with Trim Galore version 0.6.7 [39]. Quality-controlled reads were then used for genome assembly using SPAdes version 3.15.3 [40]. The STs was determined by scanning the assembled genomes against a default PubMLST typing scheme using MLST v 2.19.0 [41]. SCC*mec* Finder v. 1.2 was used with default settings to identify SCC*mec* elements [42]. We characterized the resistomes and virulence genes using ABRicate version 1.0.1. [43]. The resistome analysis was run against the CARD database with default cutoff values of 80% nucleotide identity and 80% coverage. We performed a supplemental Blast search on fluoroquinolone-resistant isolates against point mutations in the *gyrA* gene (Accession: AM262968.1) [44]. For the virulome analysis, we used an in-house database on the MRSE sequences consisting of nucleotide sequences for 27 virulence genes (Appendix A). The *S. pseudintermedius* isolates were run against the database made by Zukancik et al. [45], consisting of 69 gene sequences. The cutoff values were set to the same level as for the resistome analysis.

## Figures and Tables

**Figure 1 antibiotics-11-00637-f001:**
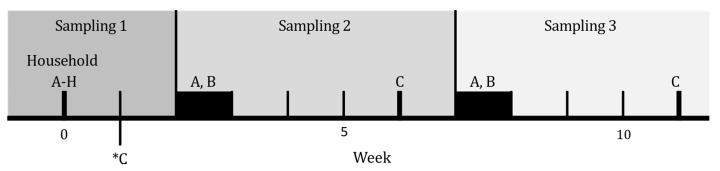
Schematic timeline of sampling days in Households A–H. All households were sampled once (Week 0). Households A and B were resampled two and seven weeks after Sampling 1. The follow-up samplings in these households lasted for five days. Household C was sampled five and 10 weeks after Dog C was euthanized (*C).

**Table 1 antibiotics-11-00637-t001:** Location of clinical methicillin-resistant *Staphylococcus* spp. (MRS) in households in Sampling 1. Contact dogs in the same household were tested if present. The contact dog of Dog C tested negative for methicillin-resistant *S. pseudintermedius* (MRSP) at the time of sampling but had tested positive for MRSP in a screening approximately one month earlier.

		Dog			Owner		Environment					Contact Dog
Household	Isolate	Infection Site	Perineum	Mouth	Nose	Throat	Food Bowl	Sleeping Place	Floor	Bathroom	Kitchen	Perineum/Mouth
A	MRSP	+	+	+	+	-	+	+	+	-	+	n/a
B	MRSP	+	-	-	-	-	-	+	+	-	-	n/a
C	MRSP	+	+	+	-	-	+	+	+	-	-	-
D	MRSP	+	+	+	-	-	+	+	+	+	+	n/a
E	MRSP	+	+	-	-	-	+	+	+	-	-	+
F	MRSP	+	+	-	-	-	+	-	-	-	-	+
G	MRSP	+	+	+	-	-	+	+	+	+	+	n/a
H	MRSE	+	+	-	+	-	+	+	+	-	+	n/a

**Table 2 antibiotics-11-00637-t002:** Phenotypic resistance in MRSP, MSSP, and MRSE in the eight households. The table presents a summary of all isolates from Sampling 1. T/S = Trimethoprim/Sulfamethoxazole, Tet = Tetracycline, Fus = Fusidic acid, Enr = Enrofloxacin, Gen = Gentamicin, Cli = Clindamycin, Oxa = Oxacillin, Cef = Cefoxitin, Chl = Chloramphenicol, Ery = Erythromycin.

Household	Isolate(s)	T/S	Tet	Fus	Enr	Gen	Cli	Oxa	Cef	Chl	Ery
A	MRSP		R				S/R	R	n/a		R
B	MRSP	R	R		R	R	R	R	n/a		R
C	MRSP	R	R		R	R	R	R	n/a		R
D	MRSP	R			R	R	R	R	n/a		R
E	MRSP	R	R					R	n/a		
F	MRSP	R	R			R	R	R	n/a		R
G	MRSP	R	R				R	R	n/a		R
G	MSSP	R	R				R		n/a		R
H	MRSE			R			S/I	R	R		R

**Table 3 antibiotics-11-00637-t003:** Summary of sequence types (ST), staphylococcal cassette chromosome *mec* (SCC*mec*) elements, and resistance genes of the MRSP and MRSE isolates isolated from Sampling 1 in all households. The ST of the MRSP isolate from household C could not be determined by multilocus sequence typing (MLST).

	Household	A	B	C	D	E	F	G	H
	Isolate	MRSP	MRSP	MRSP	MRSP	MRSP	MRSP	MRSP	MRSE
	ST	258	551	-	680	258	386	258	640
AB class	SCC*mec*	IVg (2B)	Vc (5C2&5)	V (5C2&5)	III (3A)	IVg (2B)	IVg (2B)	IVg (2B)	IVd (2B)
Aminoglycoside	*ant(6′)-la*		+	+	+	+	+	+	
	*aph(3′)-llla*		+	+	+	+	+	+	
	*aac(6′)-le*		+	+	+		+		
	*aph(2”)-la*		+	+	+		+		
	*ant(4′)-lb*								+
	*ant(9)-la*				+				
	*sat4*		+	+	+	+		+	
Beta-lactam	*blaZ*	+	+	+	+	+	+	+	+
	*mecA*	+	+	+	+	+	+	+	+
Folate pathway antagonist	*dfrG*	+	+	+	+	+	+	+	
	*dfrC*								+
Macrolide, Lincosamide, Streptogramin B	*ermB*	+	+	+	+		+	+	
	*lsaE*						+		
	*mefE*			+			+		
	*msrA*								+
Tetracycline	*tetM*	+	+	+		+	+	+	
	*tetK*		+						
Steroid antibacterial	*fusB*								+
Multidrug	*mgrA*								+
	*norA*								+

**Table 4 antibiotics-11-00637-t004:** Persistence of MRSP over time in Households A and B. Sampling Period 2 started two weeks after Sampling 1. Sampling Period 3 started four weeks after Sampling Period 2. The owners tested negative for MRSP in the follow-up sampling periods, but tested positive for MSSP (*) on two occasions each. The home environments remained positive for MRSP throughout the sampling periods.

Household A		Sampling 1	Sampling Period 2					Sampling Period 3				
		Day 1	D1	D2	D3	D4	D5	D1	D2	D3	D4	D5
Dog	Infection site	+	+		+	+	+	+	+	+	+	+
	Perineum, mouth	+	+	+					+	+		+
Owner	Nose/Throat	+	*									
Environment	Floor	+	+	+		+			+	+		
	Bathroom	+	+						+			
	Kitchen	+										
**Household B**		**Sampling 1**	**Sampling Period 2**					**Sampling Period 3**				
		**Day 1**	**D1**	**D2**	**D3**	**D4**	**D5**	**D1**	**D2**	**D3**	**D4**	**D5**
Dog	Infection site	+	+									
	Perineum, mouth								+			
Owner	Nose/Throat					*					*	
Environment	Floor	+	+	+	+	+		+	+	+	+	+
	Bathroom			+								
	Kitchen			+		+		+				

**Table 5 antibiotics-11-00637-t005:** Summary of the participating dogs. Dogs D–F were on or had received antimicrobial (AM) treatment within the past 14 days before sampling.

Dog	A	B	C	D	E	F	G	H
Breed	English Bulldog	Hungarian Vizsla	Chow Chow	English Staffordshire Bullterrier	Rottweiler	Great Dane	Bullmastiff	Rottweiler
Age	4	2	1	1	2	8 months	8	3
Sex	Neutered male	Male	Female	Male	Female	Male	Female	Male
Diagnosis	Interdigital furunculosis	Otitis externa	Pyotraumatic dermatitis	Surgical site infection	Mastitis	Surgical site infection	Surgical site infection	Pyotraumatic dermatitis
Bacteria	MRSP	MRSP	MRSP	MRSP	MRSP	MRSP	MRSP	MRSE, MSSP
Contact dog	-	-	Mixed breed (n = 1)	-	Rottweiler (n = 10)	Rottweiler (n = 1)	-	-
AM at time of sampling	-	-	-	Cefalexin	Amoxicillin Trimetho-prim	Amoxicillin Ampicillin Cefalexin Enrofloxacin	-	-

## Data Availability

The data presented in this study are available in Appendix A. Whole-genome sequence data are available at http://www.ncbi.nlm.nih.gov/bioproject/820295.

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
