# Peer review of "Transmission of Methicillin-Resistant Staphylococcus spp. from Infected Dogs to the Home Environment and Owners"

_antibiotics, 2022, doi:10.3390/antibiotics11050637_

Round 1

Reviewer 1 Report

The manuscript entitled "Transmission of methicillin-resistant Staphylococcus spp. from infected dogs to the home environment and owners " by Mari et al describes the transmission of MRS from infected dogs to their owners and home environments. The manuscript is well written and important for the dog owner and the research community involving microbiologists, veterinarians, and physicians. I recommend the manuscript to be published in Antibiotics.

Author Response

The authors thank the reviewer for reviewing our manuscript. In the resubmitted manuscript, we have added an additional reference upon request from one of the reviewers (Line 374-375). The last paragraph of the discussion in the original manuscript has been moved to the conclusions section (Line 286). In addition, we have rephrased the sentence on line 186 to clarify for the readers what proteins the bacSp222 and sec3 encode. Lastly, we have edited the sentences on lines 47 (adding "of bacteria") and 290 (adding "cleaning measures") and corrected punctuation- and grammatical errors. 

Reviewer 2 Report

Well done!  Were there people other than the owner in the households?  Presumably the "owner" was the person who had the most significant interaction with the dog.

Author Response

(The authors gave the same response as above.)

Reviewer 3 Report

The manuscript presented for review is well-written and quite simple report about the transmission of methicillin-resistant Staphylococcus spp. from infected dogs to the home environment and owners.

Methicillin resistant Staphylococcus aureus strains pose a serious treatment problem. Therefore, the subject of presented manuscript is interesting. The research was conducted in a correct way and the results are clearly presented. However, there is no statistical analysis of the results.

The manuscript is well structured, but it has some errors, and it is necessary to do a major revision to be accepted:

Line 40: in particular, are à are

Table: à Table. (Please correct it throughout)

Line 94: This reference from 10 years ago is no longer available. Please use the current definitions.

Line 372: as above

After the Material and methods chapter, a conclusions section should be added. The authors should also consider adding a simple statistical analysis.

Author Response

The authors thank the reviewer for your thorough revision of our manuscript. We appreciate your comments and believe we have improved the manuscript in accordance with your comments and advice, and these have been addressed in the resubmitted version of the manuscript. Please find our detailed point-by-point answers to your comments below (our answers in italic).

  • Line 40: in particular, are à are
  • Table: à Table. (Please correct it throughout)

The suggestions corrections have been made.

  • Line 94: This reference from 10 years ago is no longer available. Please use the current definitions.
  • Line 372: as above

As far as we know, the definition proposed by Magiorakos et al. is still used to define multidrug resistance in Staphylococcus aureus. An article by Sweeney et al. from 2018 suggested minor modifications to the definition to make it applicable to companion animal associated-staphylococci, changing the wording “non-susceptible” to “not susceptible”. Nonetheless, the modification does not affect the results in the original manuscript. We have added a citation referring to the article by Sweeney et al. in the materials and methods section (Line 375)

  • After the Material and methods chapter, a conclusions section should be added. The authors should also consider adding a simple statistical analysis.

The last paragraph of the discussion section in the original manuscript is now separated into a conclusion paragraph. According to the Antibiotics’ author guidelines, the conclusion section should be presented before the Materials and methods section. Hence, we have added the section on line 286. The article is a descriptive article based on material from eight households. Considering the small sample size, we have decided not to include any statistics in the results as we think it will be of limited value. Instead, we have focused purely on the descriptive part and presented the result in tables. 

Round 2

Reviewer 3 Report

I would like to thank the Authors for thoroughly addressing the review comments.